# The Child Behavior Checklist as a Screening Instrument for PTSD in Refugee Children

**DOI:** 10.3390/children8060521

**Published:** 2021-06-18

**Authors:** Ina Nehring, Heribert Sattel, Maesa Al-Hallak, Martin Sack, Peter Henningsen, Volker Mall, Sigrid Aberl

**Affiliations:** 1Department of Social Pediatrics, Technische Universität München, D-81377 Munich, Germany; Maesa.Al-Hallak@kbo.de (M.A.-H.); volker.mall@kbo.de (V.M.); 2Department of Psychosomatic Medicine and Psychotherapy, Klinikum rechts der Isar, Technische Universität München, D-81377 Munich, Germany; h.sattel@tum.de (H.S.); martin.sack@mri.tum.de (M.S.); p.henningsen@tum.de (P.H.); sigrid.aberl@muenchen-klinik.de (S.A.)

**Keywords:** child behavior checklist, post traumatic stress disorder, refugee, screening

## Abstract

Thousands of refugees who have entered Europe experienced threatening conditions, potentially leading to post traumatic stress disorder (PTSD), which has to be detected and treated early to avoid chronic manifestation, especially in children. We aimed to evaluate and test suitable screening tools to detect PTSD in children. Syrian refugee children aged 4–14 years were examined using the PTSD-semi-structured interview, the Kinder-DIPS, and the Child Behavior Checklist (CBCL). The latter was evaluated as a potential screening tool for PTSD using (i) the CBCL-PTSD subscale and (ii) an alternative subscale consisting of a psychometrically guided selection of items with an appropriate correlation to PTSD and a sufficient prevalence (presence in more than 20% of the cases with PTSD). For both tools we calculated sensitivity, specificity, and a receiver operating characteristic (ROC) curve. Depending on the sum score of the items, the 20-item CBCL-PTSD subscale as used in previous studies yielded a maximal sensitivity of 85% and specificity of 76%. The psychometrically guided item selection resulted in a sensitivity of 85% and a specificity of 83%. The areas under the ROC curves were the same for both tools (0.9). Both subscales may be suitable as screening instrument for PTSD in refugee children, as they reveal a high sensitivity and specificity.

## 1. Introduction

The number of refugees around the Middle East, Africa, and Afghanistan is increasing constantly, and disastrous conditions in their home countries as well as exhausting flights have an impact on the physical and psychological health of adults and children [1,2]. Hence, between 9% and 33% of adult refugees and 21–45% of children who experienced wars or armed conflicts suffer post traumatic stress disorder (PTSD) [3,4,5,6].

Psychological interventions, such as cognitive behavioral therapy, are shown to be highly effective in the treatment of traumatic symptoms [7,8] and thus may avoid chronic manifestation of the disorder. Therefore, detection of PTSD is necessary for appropriate treatment and the prevention of chronic courses; consequently, a routine screening is recommended [9]. A recent systematic review detected a lack of evidence in screening tools for PTSD in refugee children, especially below the age of six years [10]. Different self-report instruments are already in practice, such as the Child PTSD Symptom Scale, consisting of 20 items [11], the PTSD Reaction Index [12], or the Essener Trauma-Inventory for Children and Adolescents (ETI-KJ) [13]. These self-report measures, however, have been developed for children older than seven [12], eight [11], or twelve [13] years and are not appropriate for younger children. For this age group, parents or caregivers fill out the suitable checklists, such as the PTSD for Preschool Age Children [14], as well as a 15-item subscale of the Child Behavior Checklist (selection of suited items derived from the longer form [15]). Most of these instruments are specific tools for the detection of trauma induced disorders. The Child Behavior Checklist (CBCL), however, is a well-established, widely used instrument assessing not only trauma induced disorders but also the general behavioral problems of children of all age groups [15]. It is therefore an instrument that comprises a broad range of childhood psychological problems without additional strain for the patients. The CBCL, which has to be completed by parents or caregivers, is easily applicable in practice and available in more than 100 languages. A 20-item subscale of the CBCL selected by Wolfe et al. [16] covers PTSD relevant questions and might therefore be a suitable screening tool for PTSD. Previous studies already examined the use of this CBCL-PTSD subscale as a screening tool in children with traumatic experiences of various origin, such as domestic violence, sexual abuse, and unfavorable medical procedures, which may explain the different findings concerning sensitivity and specificity [15,17]. However, refugee children, according to the definition of the UNHCR (www.unhcr.org/refugees), have not yet been examined with this tool.

We therefore analyzed the data of refugee children from Syria who were hosted in a German reception camp to find out if the above mentioned CBCL-PTSD subscale might be an appropriate screening tool in this population. We further aimed to find out if a psychometrically guided item selection might be more appropriate and better tailored for the target population. Hence, we hypothesized that (i) the CBCL-PTSD subscale, as used in previous studies, would be a sensitive screening tool for PTSD in refugee children, and (ii) a psychometrically guided item selection would yield higher sensitivity and specificity in this particular population. 

## 2. Methods

A cross sectional study was conducted between January and June 2014 in a former military barracks called “Bayernkaserne”, which is used as reception camp for refugees in Munich, Germany. Data were directly collected in the barracks in separate examination rooms. The study included children with Syrian origin, aged 0 to 14 years, who were hosted in the Bayernkaserne reception camp and who were attended by at least one parent or legal guardian. Each newcomer who was listed by the agency and eligible for our study was contacted and received information about the study. They were asked to take part and invited to the first appointment. 

Informed consent was given by the parents, and children were asked if they were willing to participate. The study protocol was approved by the ethics committee of the Technical University of Munich and adheres to the Helsinki Declaration. 

The parents were interviewed and general information on age, sex, and religion, as well as subjective social status and socioeconomic status (according to MacArthur scale [18], which ranges from 1 to 10 points for low to high subjective socioeconomic status and has already been applied in immigrants) was collected. Because siblings were also included, there were less interview partners than analyzed children. For the purpose of our analyses, we extracted only children with completed psychological data who were older than 4 years. 

### 2.1. Assessment

Clinical Examination and Interview: Parents were comprehensively interviewed and children were psychologically and physically examined. Details are described elsewhere [19]. In brief, trained study personnel interviewed the parents (i.e., one parent per family) and collected general data, information on former and current medical conditions of the parents, and information on their flight. For the single parents (mother or father) who voluntarily participated, the interviewer used open questions. A pediatrician of Syrian origin performed a comprehensive physical examination of the children. To diagnose psychiatric disorders (DSM-IV) like PTSD, the children were observed and interviewed by trained child and adolescent psychiatrists who collected the case histories and applied a psychometric assessment. A psychologist monitored the developmental, behavioral, and emotional problems of the children. Further, the investigators focused on the social interaction and specific PTSD-symptoms and established a psychopathological report. Depending on the child’s age, information was received from the children or their parents. For children ≥ 6 years, the comprehensive structured Diagnostic Interview for Mental Disorders for Children and Adolescents, the Kinder-DIPS [20], was applied to establish a PTSD diagnosis according to DSM-IV. Children younger than 6 years received a clinical diagnosis from the psychologists and psychiatrists based on the German version [21] of the post-traumatic stress disorder semi-structured interview (PTSDSSI) for babies and toddlers developed by Scheeringa and Zeanah [22]. This instrument allows the diagnosis of PTSD in children older than 9 months by interviewing the main caregiver. 

Interviews were accompanied by professional translators and native speaking doctors in order to minimize language restrictions. The duration of all examinations lasted 1–2 days for one family.

### 2.2. Psychometric Testing and Statistical Analyses

The Child Behavior Checklist (CBCL), developed by T.M. Achenbach [23], is a comprehensive questionnaire that comprises 113 items on child behavior, exploring the eight problem scales describing dimensions of problematic behavior: anxiety/depression, anti-social behavior, social problems, thought problems, somatic complaints, withdrawal/depression, attention problems, and aggressive behavior. The scale allows one to draw a conclusion on the internalizing, externalizing, and overall (total) problems of the child. Answers are given on a three-point Likert-scale (0—not true, 1—somewhat or sometimes true, 2—very true or often true). The CBCL was developed to identify problems by a respondent who knows the child well, such as a parent or caregiver. We applied the full German version of the CBCL for children of 4–18 years [24]. The checklist was completed by the parents. The instrument shows high 1-week test–retest reliability (r > 0.8) [25].

### 2.3. Statistical Analysis

As selected by Wolfe et al. [16] and applied in previous studies [15,17], firstly, we likewise examined the 20-item subscale of the CBCL (CBCL-PTSD subscale), which was developed to screen for potential PTSD in children older than four years. The computation of this subscale requires a recoding of each item by pooling the “somewhat/sometimes true” and “very true/often” replies, resulting in a statement on whether a problem is present or not (i.e., bivariate data). Then, the frequencies of the condensed problematic behaviors were determined, as well as the correlation of each item with the total score (item-total correlation), as well as with the diagnostic criterion “PTSD-present” (addressing reliability and validity issues, respectively).

In the second step, we replicated these analyses with all items of the CBCL. As criteria for potentially suitable items for the prediction of possible PTSD, a significant correlation with the presence of a PTSD diagnosis (Pearson correlation r ≥ 0.2, *p* < 0.05, two-tailed) and a considerable prevalence of each particular behavior (present in at least 20% of the cases, rounded to nearest value) were chosen, resulting in a selection of 18 items from which an psychometrically guided item selection was derived [26,27,28].

To describe the sensitivity (ability to detect caseness) and specificity (precision of this detection) of this alternative and to determine possible cut-offs of both screening measures, we calculated receiver operating characteristic (ROC) curves, displaying the corresponding areas under the curve (AUC). The ROC curve graphically illustrates the suitability of a screening tool by plotting different sensitivity against specificity. An optimal cut-off considering a compromise of high sensitivity and high specificity would be chosen where the slope of the ROC-curve approaches 1, resulting in a maximum AUC of close to 1.0 [29].

Additionally, we calculated internal consistency (Cronbach’s alpha) as a measure for internal reliability and the positive predictive value (PPV). The latter is defined as the quotient of the true positive test results divided by all positive test results. A low PPV would indicate that most of the positive test results are “false positive”, suggesting a reduced test accuracy.

The analyses were performed with SPSS 23 for Windows.

## 3. Results

### 3.1. Participants

Of initially 77 eligible children, 16 (20.8%) had either no complete CBCL or no complete clinical examination and were therefore excluded. Hence, the complete data of 61 children from 38 families could be analyzed. The mean age of the children was 8.9 years (SD: 2.8); 36 (59.0%) were boys (Table 1). PTSD was diagnosed in 20 of those children (32.8%) in the clinical examination. Parents were mostly of Syrian origin, Islamic religion, and had Arabic mother tongue (Table 2).

### 3.2. Test Results

Table 3 summarizes the results for both item selections. The CBCL-PTSD subscale comprises 20 items, whereas our psychometrically guided selection comprises 18 items. Of these, eleven CBCL items overlap in the two item selections. 

#### 3.2.1. Item-Frequencies and Item/Criterion-Item/Total Correlations

(a) CBCL-PTSD subscale: The frequency of the assessed behaviors varies considerably and is observed from 2 children (3.3%, “vomiting and throwing up”) up to 29 children (47.5%, “unhappy, sad, or depressed”). Moreover, the correlation of each item with the gold standard “established PTSD-diagnosis” ranges between 0.60 (“unhappy, sad, or depressed”) to −0.04 (“nausea, feels sick”), indicating that not all items are significantly related to the presence of this diagnosis. The item-total correlation displayed concordant indices.

(b) Psychometrically guided item selection: The item frequency varies between 19.7% (*n* = 12 children with “sudden changes in moods or feelings”) and 47.5% (*n* = 29 children who were “unhappy, sad, or depressed”). The correlation between the individual items and the PTSD diagnosis ranges between 0.60 (“unhappy, sad, or depressed”) and 0.20 (“argues a lot”). Again, the item-total correlation displayed similar indices.

#### 3.2.2. Internal Consistency and Proposed Cut-Offs

(a) With a Cronbach’s α = 0.79 for the CBCL-PTSD subscale, an acceptable internal consistency of the CBCL-PTSD subscale could be observed. The maximum AUC reached 0.88. A cut-off value of 5 (representing at least five symptoms present) was associated with a sensitivity of 85% and specificity of 76%, whilst a cut-off of 7 and more symptoms present yielded a sensitivity of 75% and a specificity of 85%. Applying the higher cut-off, 24 children were tested positive, from which 17 suffered actually from PTSD according to the diagnostic interview. Accordingly, a positive predictive value (PPV) of 71.4% was determined.

(b) The psychometrically guided item selection having 18 items with a prevalence of more than 20% of the cases (Table 3) partly overlaps with the above analyzed CBCL-PTSD subscale. A Cronbach’s α of 0.89 for this selection shows a slightly improved internal consistency, as compared to the CBCL-PTSD subscale, and can be estimated to be good.

While applying this item selection, we detected an AUC of 0.86. Again, we considered two cut-offs: The optimal cut-off value of 7 (representing at least seven symptoms present) yielded a sensitivity of 85% and a specificity of 83% (Figure 1). At this cut-off we identified 17 out of 20 children with PTSD. A lower cut-off of 5 results in a sensitivity of 90% and a specificity of 73%. The positive predictive value (PPV) of 70.8% indicates that a comparable proportion of all PTSD positives were detected correctly.

## 4. Discussion

Our study examined the suitability of two CBCL subscales in a population of Syrian refugee children and preadolescents. We were able to demonstrate that both the existing PTSD-subscale of the CBCL as well as a psychometrically guided selection of items show high sensitivity and specificity for this patient group. Although we cannot detect all PTSD cases with these measures, we are not aware of a PTSD screening tool with considerably higher sensitivity and specificity. However, it is important to consider the tools for screening purposes but not for diagnosis.

The principal aim of the detection of PTSD has already been studied by Dehon and Scheeringa and by Ruggerio and McLeer [15,17], who could demonstrate a high diagnostic accuracy of the CBCL-PTSD subscale in children with different traumatic experiences. In our analyses, the sensitivity and specificity of the CBCL-PTSD subscale were 85% and 76%, whereas the study of Ruggiero and McLeer [17] yielded a sensitivity of 87% and a specificity of 61.5% and the study of Dehon and Scheeringa [15] yielded 75% and 84.4%, respectively. Moreover, both previous studies defined the optimal sum score cut-off at 8 or more symptoms present, whilst our results revealed a lower cut-off of 5 or more symptoms. A cut-off of 8 or more would, in our sample, result in a sensitivity of 0.50 only, which cannot be considered appreciable for screening purposes. As possible explanations, differences in the observed study populations are likely: Ruggiero and McLeer examined 6–16-year-old children who had been sexually abused, Dehon and Scheeringa studied preschool aged children with different types of level I trauma (such as accidents, community, or domestic violence, medical procedures, sexual abuse), whereas our study consisted of a homogenous sample of 4–14-year-old Syrian refugee children who share different experiences in Syria and on their flight to Germany. Children who have been sexually abused frequently show psychiatric disorders that are not core symptoms of PTSD, e.g., attention deficits, anxiety-depression, and social withdrawal [30,31]. Accordingly, Ruggerio and McLeer described the examined children often as affected by additional related comorbidities such as dysthymic disorder, major depressive disorders, and specific phobia. The multimorbidity of these children is likely to lead to a high sum score as a statistically optimal cut-off, combined with a high sensitivity but relatively low specificity. In contrast, we have recently shown that our sample has a relatively low rate of comorbidities apart from PTSD [19], explaining the low optimal sum score with comparable sensitivity and higher specificity.

When considering an item selection for screening purposes for the detection of PTSD, it might be helpful to target a high sensitivity, combined with reasonable specificity. Therefore, we selected items according to their psychometric properties and determined sensitivity and specificity at the same population. Even in this context, maximum sensitivity remained at 85% with slightly higher specificity (83%). Comparing the ROC curves between the existing CBCL-PTSD subscale and the subscale optimized for our population, both provided comparable detection rates. However, at the cut-off in question, where there was an increment of 10 points in sensitivity while retaining a comparable specificity, one more child out of ten could be identified as “at risk for PTSD”, with comparably small effort, when applying the psychometrically guided item selection. In general, sensitivity rates for both selections do not exceed 85% at reasonable expenses; this may point toward a ceiling effect of CBCL items concerning sensitivity for PTSD.

Regarding the individual items, those addressing sadness and depression, concentration deficits, sudden changes in mood, sleeping problems, and nightmares are strong indicators of a potential PTSD. However, some items of the CBCL-PTSD subscale, such as headaches or stomachaches, were not well correlated with the PTSD diagnosis in our population and showed a quite low item-total correlation. In particular, “physical problems without known medical cause”, items 56b,c, and g in CBCL, (e.g., headaches, nausea/ feels sick, vomiting/throwing up) were very low correlated with the PTSD diagnosis in our population. This finding is well in line with the new PTSD-criteria for children as proposed in ICD-11 [32], which consider the following clinical characteristics: disorganization, agitation, temper tantrums, clinging, excessive crying, social withdrawal, separation anxiety, distrust, trauma-specific re-enactments such as in repetitive play or drawings, frightening dreams without clear content or night terrors, sense of foreshortened future, and impulsivity. Although the majority of those criteria is covered by both item selections, a few of them are solely or more directly covered by the psychometrically guided selection (e.g., “crying a lot”). Furthermore, in contrast to the ICD-10, physical symptoms are no longer in use to specify a PTSD-diagnosis in the ICD-11. The elimination of these items from the CBCL-PTSD subscale may entail a better fit of the instrument to the forthcoming revision in ICD-11.

### Strengths & Limitations

This is the first study that examines the CBCL-PTSD subscale in a population of refugee children and preadolescents. Substantial emphasis was put on the meticulous data collection and study conduct in this sensitive and strained population. The sample is homogenous concerning the cultural background and the kind of trauma experienced at their home countries and during the flight. Furthermore, all children were accompanied by at least one parent and therefore had partly intact family. Taken together, this may be a protective factor and a reason for the low rate of psychiatric disorders apart from PTSD. Although these are favorable conditions for the validation of a screening instrument for PTSD, it is likely to limit the transfer of the results to other populations. However, in general, refugee children have similar experiences such as home loss and flight [10], whereby the results might be generalizable, at least for refugee populations.

There are some limitations, most of them due to the situation in the field. The inclusion of siblings might have biased the results toward a specific direction. However, picking out only one child per family may lead to another selection bias. Hence, we decided to also include siblings, corresponding to a real field situation. The refugees may have been scared and may have tended not to report certain behaviors or to give more desirable answers in the interview. Because we included only children with Syrian origin we cannot exclude a cultural influence when considering a behavior as appropriate or problematic, which might systematically bias the results. Although children and parents were thoroughly examined, the discussed comorbidities could not be measured comprehensively. Additionally, working together with translators—although absolutely necessary for the study conduct—may have been imprecise or even error-prone. Moreover, children with pathological anxiety may be more likely to provide socially desirable responses on self-report measures. Finally, the limited number and the specific characteristics of the participants will not allow us to generalize our findings without rigorous examination.

## 5. Conclusions

It can be concluded that the existing CBCL-PTSD subscale developed by Wolfe et al. proved to be a suitable screening tool in refugee children with high sensitivity and specificity for the detection of PTSD. Psychometric data from our study and the revised classification of PTSD in the ICD-11, however, may suggest the elimination of items referring to somatoform symptoms.

## Figures and Tables

**Figure 1 children-08-00521-f001:**
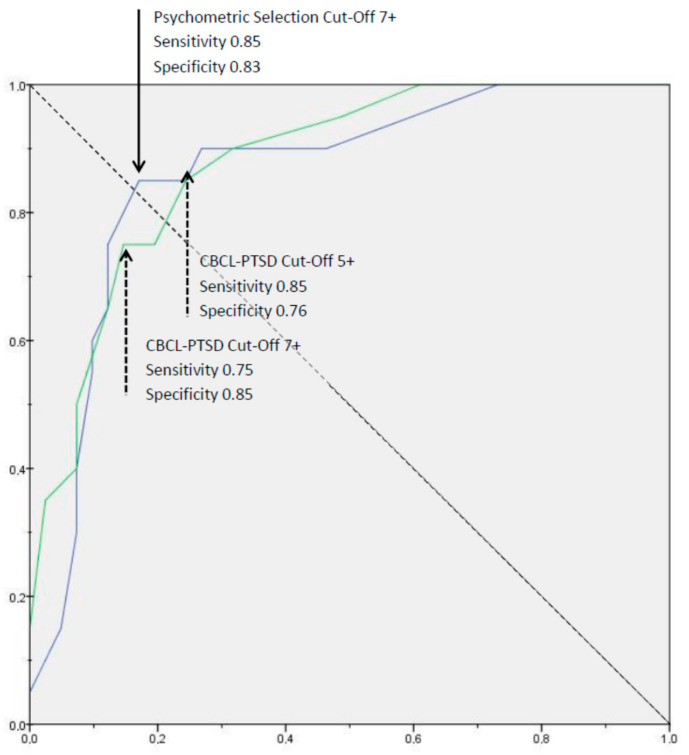
Receiver Operating Characteristic (ROC) Curve. The green line depicts the ROC curve of the PTSD-CBCL subscale (area under the curve (AUC) = 0.88), the blue line denotes the ROC curve of the psychometrically guided item selection (AUC = 0.86). “Cut-Off 7+”—at least 7 symptoms are present.

**Table 1 children-08-00521-t001:** General characteristics of the study population.

	PTSD (*n* = 20)	No PTSD (*n* = 41)
**Children 4–14 years**		
Age (years (SD))	8.2 (2.5)	9.3 (2.9)
Boys (*n*)	12	24
Religion (*n*)		
Islam	19	39
Other	1	2
In Germany since, months (SD)	1.1 (1.1)	1.1 (1.0)

SD: standard deviation.

**Table 2 children-08-00521-t002:** General characteristics of the parent who was interviewed.

	*n* (%)
**Interview partner (*n* = 38)**	
mother	27 (71.1)
father	11 (28.9)
**Country of birth**	
Syria	35 (92.1)
other (Iraq, Jordan, Libya)	3 (7.9)
**Religion**	
Islam	36 (94.7)
other	2 (5.3)
**Mother tongue**	
Arabic	30 (78.9)
Kurdish	5 (13.2)
other	3 (7.9)
Communication (language) problems in Germany ^1^	29 (76.3)
Feels socially isolated ^2^	17 (44.7)
Community-based subjective sociodemographic status in Germany above the median ^3^	19 (51.3)
Society-based subjective sociodemographic status in Germany above the median ^3^	12 (35.3)

^1^ Interview partner was asked directly if he/she had any problems to communicate in Germany. ^2^ Interview partner was directly asked if he/she feels socially isolated in the camp. ^3^ Subjective sociodemographic status according to MacArthur Scale, “above the median”—at least 5 out of 10 points on the Scale, “community-based”—in the reception camp, “society-based”—Germany in general.

**Table 3 children-08-00521-t003:** Selections of CBCL items used for analyses: CBCL-PTSD subscale as developed by Wolfe et al. [16] and psychometrically guided item selection. Eleven items (marked with ^a^) are frequent behaviors, well associated with a PTSD-diagnosis and identical in both tested item selections.

CBCL Item Selections	Coincidence of Item with PTSD Diagnosis ^1^	Occurrence/Frequency ^2^ [%]	Item—Total Correlation ^1^ (*CBCL-PTSD-Scale*)	Item—Total Correlation ^1^ (*Psycho-Metrically Guided Item Selection*)
**Frequent Behaviors, Well Associated with a PTSD-Diagnosis** **Overlapping Items of both Item Selections**				
Unhappy, sad. or depressed ^a^	0.60	47.5	0.762	0.766
Nightmares ^a^	0.48	41.0	0.597	0.702
Cannot concentrate, cannot pay attention for long ^a^	0.45	23.0	0.483	0.552
Sudden changes in mood or feelings ^a^	0.45	19.7	0.537	0.554
Trouble sleeping ^a^	0.42	36.1	0.522	0.588
Too fearful, anxious ^a^	0.39	37.7	0.451	0.507
Stubborn, sullen/irritable ^a^	0.38	34.4	0.451	0.468
Fears certain places, animals, situations other than school ^a^	0.37	35.0	0.467	0.412
Nervous, high-strung, or tense ^a^	0.33	25.0	0.455	0.480
Clings to adults or too dependent ^a^	0.22	35.0	0.381	0.394
Argues a lot ^a^	0.20	45.9	0.257	0.274
**Rare/Unassociated Behaviors** **Additional Items of CBCL-PTSD Subscale**				
Withdrawn, does not get involved with others	0.36	9.8	0.337	–
Stomachaches and cramps	0.25	13.1	0.203	–
Feels others are out to get him/her	0.25	13.1	0.305	–
Cannot get his/her mind off certain thoughts, obsessions	0.25	13.1	0.367	–
Feels too guilty	0.12	9.8	0.258	–
Vomiting and throwing up	0.07	3.3	0.061	–
Secretive and keeps things to self	0.03	27.9	0.249	–
Headaches	0.01	5.0	0.140	–
Nausea and feels sick	−0.04	6.7	0.000	–
**Additional Psychometrically Suitable Items** **(Psychometrically Guided Item Selection)**				
Disobedient at home	0.41	32.8	–	0.561
Impulsive or acts without thinking	0.32	21.3	–	0.590
Cries a lot	0.30	39.3	–	0.649
Too shy or timid	0.28	36.1	–	0.371
Does not get along with others	0.28	23.3	–	0.383
Worries	0.27	29.7	–	0.653
Can’t sit still, restless, or hyperactive	0.22	26.2	–	0.373

^a^ Items identical in both item selections. ^1^ correlation coefficient (Spearman rank correlations, rho). ^2^ “somewhat/sometimes true” or “very true/often”.

## Data Availability

Unfortunately, we cannot share the data because the approval by the ethics committee did not consider patient data publication. We are glad to be able to present the results of our analyses but are not allowed to share the original data.

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
