# Peer review of "The Child Behavior Checklist as a Screening Instrument for PTSD in Refugee Children"

_children, 2021, doi:10.3390/children8060521_

Round 1
Reviewer 1 Report
Authors could in a sentence or two elaborate on why religion and nation other than Syria believed to be important and separated out in data.
Study might be strengthened with further explication of selection choice and hypothesis of why guided selection yields higher sensitivity and selectivity. Speculation on why religion, sex, age should bear greater insight is helpful. Limitation is the very small "other" grouping for religion hurting generalizability inferences. Sample not entirely homogeneous. Lack of complete homogeneity is useful for making inferences across groupings.
Author Response
Authors could in a sentence or two elaborate on why religion and nation other than Syria believed to be important and separated out in data.
Answer: Thank you for raising this issue. We focussed on children with Syrian origin because this is a homogeneous group which has been well characterized in previous studies. However, this may limit our results. We therefore added to the discussion section: “Since we included only children with Syrian origin we cannot exclude a cultural influence when considering a behavior as appropriate or problematic which might systematically bias the results.”
Study might be strengthened with further explication of selection choice and hypothesis of why guided selection yields higher sensitivity and selectivity. Speculation on why religion, sex, age should bear greater insight is helpful. Limitation is the very small "other" grouping for religion hurting generalizability inferences. Sample not entirely homogeneous. Lack of complete homogeneity is useful for making inferences across groupings.
Answer: It is just an assumption that a computed item selection might be better tailored and therefore yield higher sensitivity and specificity.
As written above, we examined Syrian refugee children only to reveal a heterogeneous group. We also mention this in the discussion section.
Reviewer 2 Report
This is a nice study with the objective to evaluate and test suitable screening tools to detect PTSD in children refugees. The authors were able to demonstrate that both, the existing PTSD-subscale of the CBCL, as well as a psychometrically guided selection of items showed high sensitivity and specificity for this children group.
The structure of the study in the appropriate sections is excellent. Vocabulary is concise and scientific. While, almost all points are supported by current and appropriate references.
Some points need minor revisions and more clarity
Line 35 I would suggest chronic course instead of chronic diseases
Line 44 better use ..”selection of suited items derived from the longer form”
Line 49 “The CBCL is easily applicable in the practice and available in more than 100 languages…” need a reference support.
Line 53 It should be clarified that the different results of sensitivity and specificity are due to the different origins of traumatic events examined in these studies, and not because the overall studies were few.
Line 81 I am not sure about the extracted meaning regarding the siblings...
Line 83 “..with completed psychological data.”
Line 76. It’s essential to add: Were all procedures based to the 1975 Helsinki Declaration, as revised in 2008?
Line 90: I would rephrase “...For the only parent (mother or father) involved, the interviewer used open questions ..”
Line 275 It would be noteworthy to mention, as a limitation, that children with pathological anxiety may be more likely to provide socially desirable responses on self-report measures.
Author Response
Some points need minor revisions and more clarity
Line 35 I would suggest chronic course instead of chronic diseases
Answer: Ok, we changed the wording.
Line 44 better use ..”selection of suited items derived from the longer form”
Answer: Thanks. We added “derived”.
Line 49 “The CBCL is easily applicable in the practice and available in more than 100 languages…” need a reference support.
Answer: OK.
Line 53 It should be clarified that the different results of sensitivity and specificity are due to the different origins of traumatic events examined in these studies, and not because the overall studies were few.
Answer: We deleted “few” and talk about “previous studies” now.
Line 81 I am not sure about the extracted meaning regarding the siblings...
Answer: There was at least one child per interview partner (normally per parent) because we included siblings in our analyses.
Line 83 “..with completed psychological data.”
Answer: Thanks.
Line 76. It’s essential to add: Were all procedures based to the 1975 Helsinki Declaration, as revised in 2008?
Answer: We added that the study adheres to Helsinki declaration.
Line 90: I would rephrase “...For the only parent (mother or father) involved, the interviewer used open questions ..”
Answer: Thanks, we rephrased the sentence accordingly.
Line 275 It would be noteworthy to mention, as a limitation, that children with pathological anxiety may be more likely to provide socially desirable responses on self-report measures.
Answer: Thank you for this hint. We added a sentence.
Reviewer 3 Report
Very well written manuscript.
Author Response
Very well written manuscript.
Thank you.